# Frequency of Dynamic Fire Behaviours in Australian Forest Environments

**Alexander I. Filkov** [1,2,*] **, Thomas J. Duff** [1] **and Trent D. Penman** [1]

[1] School of Ecosystem and Forest Sciences, Faculty of Science, University of Melbourne, Creswick, Victoria 3363, Australia; tjduff@unimelb.edu.au (T.J.D.); trent.penman@unimelb.edu.au (T.D.P.)
[2] Bushfire and Natural Hazards Cooperative Research Centre, Melbourne, Victoria 3002, Australia
[*] Correspondence: alexander.filkov@unimelb.edu.au; Tel.: +61-3-5321-4198

**Abstract:** Wildfires can result in significant social, environmental and economic losses. Fires in which dynamic fire behaviours (DFBs) occur contribute disproportionately to damage statistics. Little quantitative data on the frequency at which DFBs occur exists. To address this problem, we conducted a structured survey using staff from fire and land management agencies in Australia regarding their experiences with DFBs. Staff were asked which, if any, DFBs were observed within fires greater than 1000 ha from the period 2006–2016 that they had experience with. They were also asked about the nature of evidence to support these observations. One hundred thirteen fires were identified. Eighty of them had between one and seven DFBs with 73% (58 fires) having multiple types of DFBs. Most DFBs could commonly be identified through direct data, suggesting an empirical analysis of these phenomena should be possible. Spotting, crown fires and pyro-convective events were the most common DFBs (66%); when combined with eruptive fires and conflagrations, these DFBs comprise 89% of all cases with DFBs. Further research should be focused on these DFBs due to their high frequencies and the fact that quantitative data are likely to be available.

**Keywords:** wildfire; dynamic fire behaviours; direct data; indirect data; anecdotal data

## 1. Introduction

Extreme wildfires create disproportionate risks to environmental and human assets. Fire propagation can be significantly affected by dynamic feedback processes that result in unpredictable behaviour, and the continual escalation of fire spread rates and intensities even when environmental conditions are consistent. These fires behave in a manner that goes beyond the suppression means and fire-fighters are unable to control the fire spread even in the most prepared and equipped regions [1]. The erratic behaviour and difficulty to control mean that these fires can burn larger areas and result in the loss of life. Throughout the world, such wildfires have occurred almost every year for the past 15 years [2–13]. In 2017 alone, 11 people were killed in Chile [14]; 109 people were killed in Portugal [8,15] and 42 people were killed in California, USA [16]. The trend for the occurrence of extreme wildfires appears to be increasing each year [8,9,17,18].

Extreme fire behaviours are physical phenomena that can occur within any fires [19–23]. These phenomena can influence the intensity, rate of growth and impact of wildfires [24–27]. Although there are studies describing such behaviours and processes behind them, the understanding of the drivers of extreme fire behaviours (EFBs) and their occurrence is still limited. In part, this may be due to the challenges with observing rare complex phenomena that occur under dangerous conditions. On the other hand, a lack of a clear definition and understanding of what the EFBs are makes it difficult to formalise methods to collect information about them.

The term "*extreme fire behaviour*" was introduced for the first time by Byram in his 1954 paper [28]. It was presented in the context of "blowup" fire, "which suddenly, and often unexpectedly, multiplies its rate of energy output many times." He did not define EFB but introduced it as a list of known conditions ("the facts of fire behaviour") associated with blowups. Most of the facts are associated with fire intensity and its disproportional increase and this makes it very challenging to draw a line between a blowup fire and EFB. Since then, the term EFB has been used extensively.

In the last decade, several attempts have been made to develop a definition of EFB. The National Wildfire Coordinating Group (https://www.nwcg.gov/) uses the following definition:

> "Extreme implies a level of fire behavior characteristics that ordinarily precludes methods of direct control action. One or more of the following is usually involved: high rate of spread, prolific crowning and/or spotting, presence of fire whirls, strong convection column. Predictability is difficult because such fires often exercise some degree of influence on their environment and behave erratically, sometimes dangerously."

This definition is similar to Byram's definition [28] and is based on a level of fire intensity and suppression. According to Werth, et al. [21] it is "intractable for scientific purposes." An alternative definition was produced as an outcome of an examination of extreme fire behaviours by a group of researchers [20,21]: "Fire spread other than steady surface spread, especially when it involves rapid increases." They highlighted the unsteady/dynamic nature of EFB. Also, the proposed definition omitted a level of fire intensity and fire suppression component. Viegas [19] proposed a definition that is similar to the National Wildfire Coordinating Group's definition: "Extreme Fire Behaviour is the set of forest fire spread characteristics and properties that preclude the possibility of controlling it safely using available present day technical resources and knowledge." The word "extreme" in the term "extreme fire behaviour" he associated with the local or absolute maximum of a fire line intensity (the rate of thermal energy released by the fire front per unit of length and of time, W/m) for which fire control is not possible. Although he mentioned that it is not possible with any degree of certainty to say what is the absolute maximum value of extreme fire behaviour and it will change in the future, he suggested using 10 MW/m as an absolute maximum. He also introduced a concept of "dynamic fire behaviour", as the fourth dimension of fire spread additionally to fuel, topography, and meteorology. Although other studies did not propose a definition of EFB, the term EFB was used extensively to describe different forms of EFB. For instance, "can involve significant and rapid changes in fire behaviour without significant changes in ambient conditions" [29], "abrupt changes in fire behaviour characteristics" [30], "may induce very high values of the rate of spread" [31], etc.

Definitional issues are present for both EFBs and extreme fires. Tedim et al. [32] (p. 5) proposed the term "*extreme fire event*" instead of "extreme fire". Based on a comprehensive literature review Tedim et al. [32] described an extreme fire event as a combination of EFBs and the consequences of them. However, they only considered a limited number of EFBs. There is still some confusion regarding extreme fire events and extreme fire behaviours. Some researchers perceive them as the same event/process and some the complete opposite (between them). This reflects in the appearance of fire intensity and suppression components in the definition of extreme fire behaviour. According to Collins' dictionary [33] the term "extreme" is used "to describe situations and behaviour which are much more severe or unusual than you would expect." It means that it always compares with normal/usual fire behaviour, which varies for different parts of the world and people's understanding [19]. This is a subjective approach; for instance, fire conditions change all time, and what would be perceived as extreme 10 years ago may not be perceived as extreme now. Also, EFBs are not always extreme, but they can be a precursor to extreme fire events. Additionally, the term "extreme" does not reflect the dynamic or unsteady nature of these phenomena and rapid changes in fire behaviour. Recently, a wildfire community started to use the term "dynamic fire behaviour" to describe different unsteady fire behaviours [29,30,34]. It means that DFB is not a single process or event combining different phenomena, but a unique physical phenomenon having an unsteady/dynamic nature. Based

on the information stated above we propose using the term "*dynamic fire behaviour*" (DFB hereafter) rather than "extreme fire behaviour". In this paper, we define DFB as a

> "physical phenomenon of fire behaviour that involves rapid changes of fire behaviour and occurs under specific conditions [19] which has the potential to be identified, described and modelled."

In general, extreme fire (or an extreme fire event) can involve one to several DFBs simultaneously [24,32]. To be consistent we will use hereafter the term DFB for all previous studies where it was mentioned as EFB.

Key DFBs were documented in the 1950s [21] and were described by Byram [28] as "blowup," "conflagration," and "erratic" behaviours that occurred in "unusual high-intensity fires." Attempts have been made to summarise the state of knowledge about DFBs [20,21,35], with the first detailed description and categorisation of DFBs by Viegas in 2012 [19].

Several recent studies have been devoted to describing the processes behind particular DFBs [25,26,36–39]. However, there is no quantitative research describing the frequency of DFBs. The DFBs that are common and have substantial impacts on fire behaviour should be prioritised for the development of models so that their physical processes can be understood and they can be predicted for operational fire management purposes. The factors that cause different DFBs cannot be statistically analysed without replications of observations from wildfires, as they are likely to be the result of a complex interaction across many parameters, such as weather, terrain and the fire itself. To understand the importance of DFBs in fire behaviour, we initially need to understand how frequently they occur to prioritise future research efforts.

Expert elicitation approaches can be used when existing data and models cannot provide sufficient data [40–43]. These approaches present challenges as they are applied when empirical data are not available and thus evaluating the accuracy is difficult [44]. However, expert elicitation can make a valuable contribution to informed decision-making in the absence of data. Here we use an expert elicitation approach to determine the relative frequency of occurrence of nine recognised manifestations of dynamic fire behaviours: spotting, crown fires, pyro-convective events (PyroEvs), eruptive fires, conflagrations, junction fires, fire tornados/whirls, fire channelling and downbursts.

## 2. Materials and Methods

DFBs have been reported to be a feature of extreme fires. To collect data on these, we considered all fires greater than 1000 ha in Australia that occurred between 2006 and 2016. The fires occurred primarily in the following vegetation types [45]: eucalypt open forests (48%); mallee woodlands and shrublands (16%); eucalypt woodlands (15%); eucalypt open woodlands (5%); heathlands (4%); other grasslands, herblands, sedgelands and rushlands (4%); eucalypt tall open forests (3%); other shrublands (1%). Across the study area and study period most jurisdictions in Australia have experienced a number of fires with DFBs (e.g., [24]) across a range of weather scenarios.

We approached 21 representatives from management agencies responsible for fire response in each state (in Australia forest and fire management is predominantly done at a state level). Sample size estimation (https://www.abs.gov.au/websitedbs/D3310114.nsf/home/Sample+Size+Calculator) showed that a minimum of 9 representatives are required to get a 2.6% relative standard error with 95% confidence level, a population size of 21, the proportion of the population to have required attribute of 0.99 (as experts survey only familiar fires) and a confidence interval of 0.05.

Experts represented different divisions and branches (e.g., operations, management, planning) and had a good understanding of bushfires in their region. Specifically, individuals were assessed as being suitable experts against the following criteria: eyewitness/familiar with surveying bushfires, firefighting or management knowledge/experience in the relevant state, a general understanding of the bushfires and DFBs, and a knowledge/understanding of planning approaches. Once a suitable expert was identified, they were contacted via email and then with a follow-up telephone call. During the telephone call, experts were provided with detailed instructions regarding the survey. Experts were

asked only to provide data on DFBs for fires for which they were familiar. Definitions for DFBs were discussed during the telephone call to ensure all experts were working to the same definitions. A presentation containing the definitions and examples was also sent to the experts via email and participants were asked to view the presentation before completing the survey. The survey asked which (if any) DFBs had been observed or not in the fires they were familiar with and what was the nature of the data to support this. Experts were asked to categorise data into three types: direct measurements (linescans, images, video, etc.), indirect data (weather records, etc.) and the data based on anecdotal evidence (observations recorded in situation reports, etc.).

Experts were only provided with fires for their jurisdiction to simplify the survey sheet and ensure answers were only provided for fires they were potentially familiar with. DFBs were classed into nine different types:

**Spotting**. Spotting is a "behaviour of a fire producing firebrands or embers that are carried by the wind and which start new fires beyond the zone of direct ignition by the main fire" [46].

**Fire tornados**. A fire tornado/whirl is a "spinning vortex column of ascending hot air and gases rising from a fire and carrying aloft smoke, debris, and flame. Fire whirls range in size from less than one foot (0.3 m) to over 500 feet (152 m) in diameter. Large fire whirls have the intensity of a small tornado" [46].

**Fire channelling**. Fire channelling/lateral vortices is a rapid lateral fire spread across a steep leeward slope in a direction approximately transverse to the prevailing winds [39].

**Junction fires**. Junction fires/junction zones (jump fires previously) are associated with the merging of the fire fronts producing very high rates of spread and with the potential to generate fire whirls and tornadoes [19].

**Eruptive fires**. Eruptive fires are fires that occur usually in canyons or steep slopes and are characterised by a rapid acceleration of the head fire rate of spread [19].

**Crown fires**. Van Wagner [47] recognized three types of crown fires according to their degree of dependence on the surface fire phase: passive, active, and independent. In our study we refer to active and independent crown fires as dynamic fire behaviours. Active crown fire is "a fire in which a solid flame develops in the crowns of trees, but the surface and crown phases advance as a linked unit dependent on each other" [46]. Independent crown fires "advance in the tree crowns alone, not requiring any energy from the surface fire to sustain combustion or movement" [46].

**Conflagrations**. "Conflagrations are raging, destructive fires. Often used to connote such a fire with a moving front as distinguished from a fire storm" [46].

**Downbursts**. Downbursts are downdrafts associated with cumulus flammagenitus clouds [48] (cumulus flammagenitus is also known by the unofficial, common name "pyrocumulus" [48]. Cumulus (Cu) are detached clouds, generally dense and with sharp outlines, developing vertically in the form of rising mounds, domes or towers, of which the bulging upper part often resembles a cauliflower. The sunlit parts of these clouds are mostly brilliant white; their bases are relatively dark and nearly horizontal [48]. Flammagenitus (Fg) are clouds that are observed to have originated as a consequence of localized natural heat sources, such as forest fires, wildfires or volcanic activity and which, at least in part, consist of water drops (for example, cumulus flammagenitus (CuFg) or cumulonimbus flammagenitus (CbFg)) [48]. Cumulonimbus (Cb) are heavy and dense clouds, with a considerable vertical extent, in the form of a mountain or huge towers. At least part of their upper portion is usually smooth, or fibrous or striated, and nearly always flattened; this part often spreads out in the shape of an anvil or vast plume [48]) that induce an outburst of strong winds on or near the ground [49]. These winds spread from the location of the downbursts and may result in a fire spread contrary to the prevailing wind direction.

**Pyro-convective events**. A pyro-convective event is an extreme manifestation of a flammagenitus cloud, generated by the heat of a wildfire, that often rises to the upper troposphere or lower stratosphere [48].

Responses were grouped for each fire. Where multiple agencies responded with information for the same fire, the data from the highest quality data type was retained. Data were analysed regarding the relative frequency of DFBs, the quantity of DFBs per fire, and the confidence level of the data.

The relative frequency $f_i = n_i/N$ (or empirical probability) of an event was used in our study. It was calculated as the absolute frequency $n_i$ normalized by the total number of events $N$. The values $f_i$ for all events $i$ were plotted to produce a frequency distribution.

## 3. Results and Discussion

Responses were received from five experts from New South Wales (NSW, Table A1), four from Victoria (VIC, Table A2), and one each from South Australia (SA, Table A3) and Tasmania (TAS, Table A4). A list of 934 fires was used for surveying, where 471 fires were from NSW, 130 from VIC, 281 from SA and 55 from TAS. No responses were received from the Australian Capital Territory (ACT), Western Australia (WA), Queensland (QLD), and the Northern Territory (NT). Information on DFBs (or absence of them) was received for a total of 113 fires (Table A1, Appendix A). Each expert provided a unique list of fires with or without DFBs. The sample size of 113 fires provided 8.9% relative standard error with the following criteria: 95% confidence level, population size 934, and proportion of the population 0.5 (a conservative estimate of variance) (https://www.abs.gov.au/websitedbs/D3310114.nsf/home/Sample+Size+Calculator).

More than half of the fires considered in the survey had at least one DFB (overall 60%). This value was consistent between states, with NSW having DFBs in 62% of fires, VIC 61%, SA 85% and TAS 57%. The occurrence of DFBs versus fire size is shown in Figure 1.

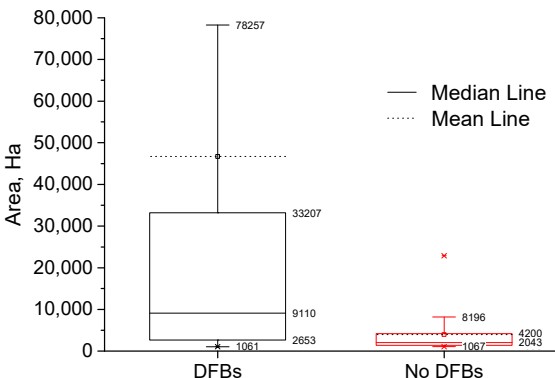

**Figure 1.** Size of fires with and without dynamic fire behaviours (DFBs).

There was a weak and approaching-borderline statistical significance relationship between occurrence of DFBs and fire size. The Pearson's correlation coefficient, R square and significance were r = −0.16, $R^2$ = 0.026 and $p$ = 0.079 respectively. Figure 1 shows that 50% of fires with DFBs had an area between 2653 and 33,207 Ha, with the median 9110 Ha.

All DFBs were recorded at least four times with crown fires and spotting being observed most frequently (60 and 59 times respectively). Table 1 shows that the fire tornado/whirls (n = 5), fire channelling (n = 4) and downburst (n = 5) were observed the fewest times.

The relative frequency of various DFBs is presented in Figure 2. Figure 2a shows the percentage of occurrence of each DFB form per fire. Spotting and crown fires were the most frequent DFBs, making up a total of 50% of all DFB observations. PyroEvs, eruptive fires and conflagrations were observed to have similar frequencies of occurrence, accounting for 39% of the remaining observations. Junction fires, fire tornado/whirls, fire channelling and downbursts combined accounted for 11% of DFBs in total.

**Table 1.** Dynamic fire behaviours. Tally of dynamic fire behaviours in depends on the data type.

| Data Type | Spotting | Fire Tornado/Whirls | Fire Channelling | Junction Fires | Eruptive Fires | Crown Fires | Confla-Grations | Down-Bursts | PyroEvs | Total |
|---|---|---|---|---|---|---|---|---|---|---|
| Direct | 27 | 3 | 2 | 4 | 13 | 22 | 14 | 2 | 27 | *114* |
| Indirect | 14 | 0 | 1 | 7 | 13 | 20 | 4 | 2 | 5 | *66* |
| Anecdotal | 18 | 2 | 1 | 1 | 4 | 18 | 6 | 1 | 4 | *55* |
| Total | *59* | *5* | *4* | *12* | *30* | *60* | *24* | *5* | *36* | *235* |

The low frequency of last four DFBs may be connected with limited knowledge about them, challenges of identification and limited data. For example, fire channelling was only described in 2012 [39]. The detection of downbursts requires local measurements of weather, which, given the sophistication of equipment required, is rare. Scale effects and the transience of events also could be mean that the frequencies of observation do not reflect the frequency of occurrence. Some DFBs occur only at large scales, e.g., PyroEvs and conflagrations, whereas junction fires and fire whirls can manifest at smaller scales and may only occur for seconds or minutes. Sometimes, the same DFB can manifest at different scales, e.g., spotting can be classified into three categories, depending on the distance and the distribution density: short distance spot fires (up to 750 m), average distance spot fires (1000–1500 m) and long distance spot fires (>5000 m) [24]. All of these make the identification of DFBs a very challenging task.

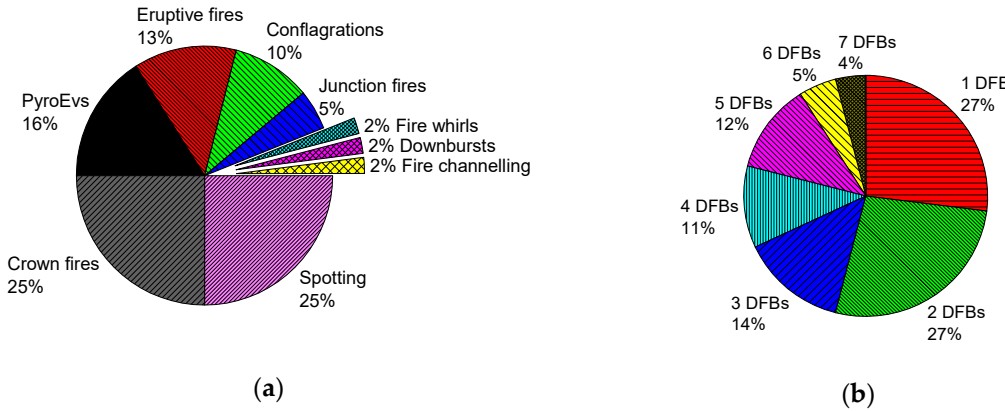

(**a**)                  (**b**)

**Figure 2.** Relative frequency of dynamic fire behaviours (DFBs). Figure (**a**) shows the relative frequency of each DFB form. The sum of all DFBs is 100%. Figure (**b**) shows percentage of fires with different quantities of different DFBs.

Spotting, crown fires, PyroEvs, eruptive fires and conflagrations were the most frequent DFBs observed. They can be more easily identified and detected, and fire managers are more likely to be familiar with them. Crown fires in conifer forests have been studied in detail, with several empirical models developed [50]. More recently, attention has been given to other DFBs, in particular PyroEvs [25,51], spotting [52,53], fire channelling [39] and fire whirls [38,54]. Most of these studies are based on Computational Fluid Dynamics or conceptual modelling and they are not experimentally validated. Results from these studies cannot be translated into systems for prediction during fires for operational decision support at this point. Conflagrations and downbursts have not been included in any physical or operational models to date.

Eighty fires had at least one DFB observed (Figure 2b). Two and more DFBs were recorded in 73% of these eighty fires. Therefore, their interactions could have complementary effects on fire behaviour, e.g., crown fires and PyroEvs could facilitate long distance spotting and fire tornados/whirls. Similar percentage of spotting and crown fires (Figure 2a) can be indirect proof of this assumption. Consequently, the potential interactions of these phenomena should be a focus of investigation.

Relationship between DFBs and their type is presented in Figure 3.

A weak positive linear relationship between the number of DFBs per fire and fire size was observed (Figure 3a) (Pearson's correlation coefficient r = 0.25, $R^2$ = 0.07) and it was statistically significant (*p* = 0.014). This means that DFBs can happen at any scale and even smaller fires (left part of area distribution on Figure 3a) can have multiple DFBs.

Analysis of Figure 3b showed that DFBs can emerge at any fire scale and fire area does not limit manifestation of DFBs. The only exception was observed for fire channelling and downbursts, but as discussed above this may be a function of limited knowledge and identification challenges.

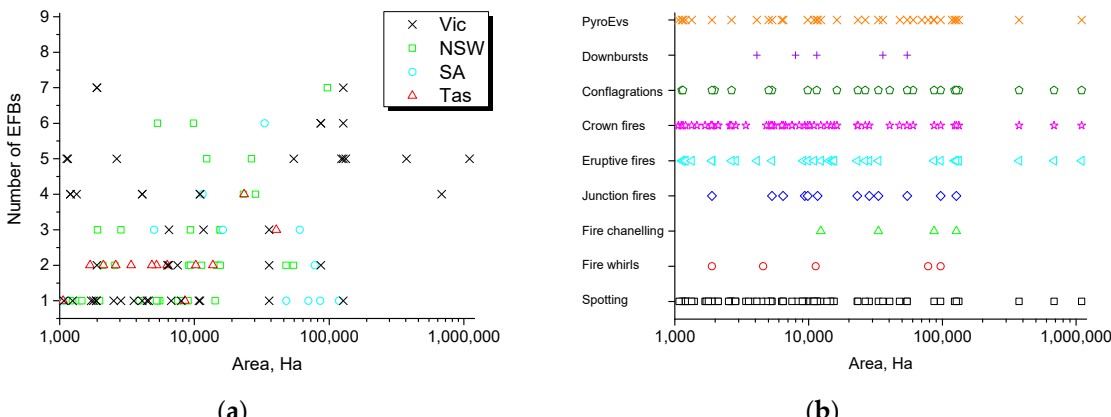

**Figure 3.** Number of dynamic fire behaviours (DFBs) and their type versus fire area. Figure (**a**) shows number of DFBs for four states and Figure (**b**) the manifestation of different types of DFBs.

Roughly half of the observations of DFBs in eighty fires were recorded as direct data, with a 49% average across all DFBs (Figure 4a). Indirect and anecdotal data accounted for similar proportions (28% and 23%, respectively). The highest percentage of DFBs supported with direct data was for PyroEvs, conflagrations, fire tornado/whirls and fire channelling, with direct data in over 50% of cases. However, due to the limited number of observations for fire tornado/whirls (5 cases) and fire channelling (4 cases), these results could be overestimated. The percentage of direct data for all DFBs was always higher than anecdotal data. Despite this, there have been few studies devoted to analysis of DFBs. The number of events where DFBs are supported by direct data indicate that there is potential for future quantitative studies. While the indirect data varied between DFBs, the highest percentage of indirect data is observed for eruptive fires, junction fires and downbursts (>40%). As noted above, the small spatial and temporal scales of these DFBs makes it difficult to collect direct data.

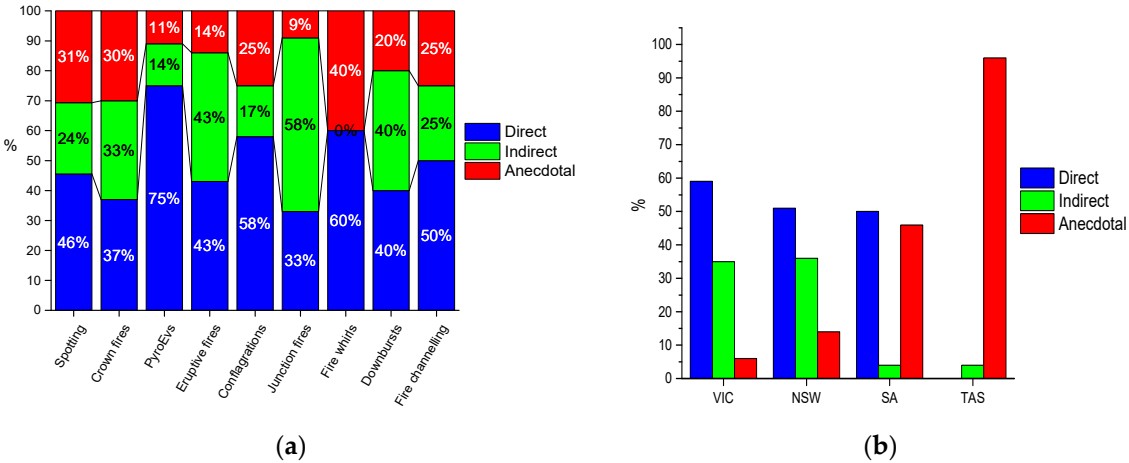

**Figure 4.** Comparison of dynamic fire behaviours (DFBs) distributions for different data types (**a**) and data types among the states (**b**), where VIC, NSW, SA, TAS are Victoria, New South Wales, South Australia, and Tasmania, respectively. The sum of fires supported by direct, indirect, and anecdotal data for each DFB is equal to 100%.

Analysis of data types between the states (Figure 4b) shows that VIC and NSW have similar patterns in the way DFBs are observed, with a high number of total observations and a high proportion of these being in direct data. This is likely to be reflective of the higher frequency of damaging wildfires in these states, which has resulted in a greater investment in infrastructure for fire monitoring. The greater availability of information for these states means that they are potentially suited for the future collection of quantitative data for the robust analysis of DFBs. Such data are important, as while

there has been work in emulating the physical processes of DFBs, there has been limited opportunity to confirm these processes with field measured data.

It should be noted that the obtained values are not the definitive values, as the limited amount of data restricts quantitative analysis. Several attempts were made to increase the database, but despite sustained efforts we were unable to increase it substantially. However, obtained values provide useful insights into frequencies of DFBs and can be used for the prioritisation of future research.

## 4. Conclusions

There is no consensus in the literature on what dynamic fire behaviour (extreme fire behaviour previously) is and no final list of their forms and agreed definitions. Rather than pursuing semantic arguments, more effort is required to understand, describe and utilize DFBs.

We found that DFBs frequently occur (71% of studied fires) and often with multiple DFBs per fire. Predictions of fire impacts will be less accurate until these phenomena are considered. Our survey indicated that spotting, crown fires, PyroEvs, eruptive fires and conflagrations are the most commonly observed DFBs, and so these should be the highest priority in determining which DFBs to research for inclusion in fire models. The relative commonness of direct evidence available for DFBs is indicative that there should be data available for the development of models. All the DFBs considered can take place in any landscape. Therefore, despite data having been collected for Australian fires, the approach demonstrated here on collecting information about DFBs is likely to be applicable for qualitative analysis around the world.

**Author Contributions:** Conceptualization, T.D.P., T.J.D. and A.I.F.; methodology, T.D.P. and A.I.F.; formal analysis, T.D.P., T.J.D. and A.I.F.; investigation, A.I.F.; writing—original draft preparation, A.I.F.; writing—review and editing, T.D.P. and T.J.D.; visualization, A.F. All authors have read and agreed to the published version of the manuscript.

**Funding:** This research was funded by the Bushfire and Natural Hazards Cooperative Research Centre 'Determining threshold conditions for extreme fire behaviour' project.

**Acknowledgments:** The work was funded by the Bushfire and Natural Hazards Cooperative Research Centre. Thomas Duff is supported by funding from the Victorian Department of Environment, Land, Water and Planning. We wish to acknowledge the agencies and individuals that participated in the study.

**Conflicts of Interest:** The authors declare no conflict of interest. The founding sponsors had no role in the design of the study; in the collection, analyses, or interpretation of data; in the writing of the manuscript, and in the decision to publish the results.

## Appendix A

Data collection survey. Dynamic fire behaviours (DFBs) observed by the experts (or their absence) and the nature of the data to support this.

Table A1. New South Wales.

| Fire Name | Fire Date (yyyy/mm/dd) | Fire Area, Ha | Dynamic Fire Behaviours | | | | | | | | |
|---|---|---|---|---|---|---|---|---|---|---|---|
| | | | Spotting | Fire Tornado/Whirls | Fire Channelling | Junction Fires | Eruptive Fires | Crown Fires | Confla-Grations | Down-Bursts | PyroEvs |
| Belimbla | 2009/02/07 | 1206 | | | | | | | | | |
| Beulah | 2013/01/20 | 1982 | | | | | | | 1 | | |
| Billo Road | 2006/12/10 | 11,302 | 3 | 3 | | | | | | | |
| Blackjack Mountain | 2013/10/14 | 1118 | | | | | | 2 | | | |
| Bulli Tops | 2012/08/10 | 1395 | 1 | | | | | | | | |
| Bush Alight-Sawn Rocks | 2009/11/17 | 5547 | | | | | | 2 | | | |
| Cobbler Road | 2013/01/08 | 14,248 | | | | | 1 | | | | |
| Curricabark Tops | 2012/09/29 | 5340 | | | | | | 2 | | | |
| Dangar Road | 2015/03/04 | 15,063 | 1 | | | | 1 | | | | |
| Deans Gap | 2013/01/07 | 9110 | 1 | | | | 3 | | | | |
| Dromedary | 2009/08/27 | 3886 | | | | | | | | | |
| Garnpang | 2012/09/28 | 7528 | | | | | | 1 | | | |
| Gibsons Plantation | 2012/10/08 | 8196 | | | | | | | | | |
| Gold Mine Road Yambulla | 2015/12/19 | 2851 | 1 | | | | 1 | 1 | | | |
| Goldmine Road | 2006/12/14 | 28,381 | 2 | | | 2 | 2 | 2 | | | |
| Goonoo Frost Road | 2007/01/12 | 26,500 | 1 | | | | 1 | 1 | 1 | | 1 |
| Hall Road, Balmoral (expert 1) | 2013/10/17 | 15,544 | 1 | | | | 1 | 2 | | | |
| Hall Road, Balmoral (expert 2) | 2013/10/17 | 15,544 | 1 | | | | 3 | | | | |
| Hank Street | 2013/10/13 | 5238 | | | | | | | | | |
| Hungerford Creek | 2013/10/14 | 47,850 | 1 | | | | | 1 | | | |
| Jingera Rock | 2009/01/24 | 2575 | 1 | | | | | 1 | | | |
| Kerringle | 2006/11/29 | 23,107 | 2 | | | 2 | 2 | 2 | | | |
| Lawsons Long Alley | 2006/11/14 | 14,617 | 3 | | | | | 3 | | | |
| Lower Mangrove | 2007/01/21 | 1911 | 3 | | | | 3 | 3 | | | |
| Mt Tangory | 2013/10/13 | 1464 | | | | | | 2 | | | |
| Mt York Rd (expert 1) | 2013/10/17 | 9358 | 3 | | | | | 3 | | | |
| Mt York Rd (expert 2) | 2013/10/17 | 9358 | 3 | | | 3 | | 3 | | | |
| Pilliga | 2006/11/29 | 97,042 | 1 | 1 | | 2 | 1 | 1 | 1 | | 1 |

**Table A1.** *Cont.*

| Fire Name | Fire Date (yyyy/mm/dd) | Fire Area, Ha | Dynamic Fire Behaviours | | | | | | | | |
|---|---|---|---|---|---|---|---|---|---|---|---|
| | | | Spotting | Fire Tornado/Whirls | Fire Channelling | Junction Fires | Eruptive Fires | Crown Fires | Confla-Grations | Down-Bursts | PyroEvs |
| State Mine (expert 1) | 2013/10/16 | 54,327 | 3 | | | | | 3 | | | |
| State Mine (expert 2) | 2013/10/16 | 54,327 | 3 | | | | | 3 | | | |
| Terraborra North | 2015/12/10 | 5325 | 1 | | | 1 | 1 | 1 | 1 | | 1 |
| Timor Road | 2006/11/29 | 9848 | 2 | | | 2 | 2 | 2 | 2 | | 2 |
| Tinkrameanah | 2006/11/28 | 2097 | 2 | | | | | 2 | | | |
| Wadbilliga | 2009/07/18 | 1101 | | | | | | | | | |
| Wirritin (expert 1) | 2013/10/16 | 8976 | | | | | | | | | |
| Wirritin (expert 2) | 2013/10/16 | 8976 | 1 | | | | | | | | |
| Yarrabin | 2013/01/06 | 12,343 | 1 | | 1 | | 1 | 1 | | | 1 |

1 is direct measurements (linescans, images, video, etc.), 2 is indirect data (weather records, etc.) and 3 is the data based on anecdotal evidence (observations recorded in situation reports, etc.).

**Table A2.** Victoria.

| Fire Name | Fire Date (yyyy/mm/dd) | Fire Area, Ha | Dynamic Fire Behaviours | | | | | | | | |
|---|---|---|---|---|---|---|---|---|---|---|---|
| | | | Spotting | Fire Tornado/Whirls | Fire Channelling | Junction Fires | Eruptive Fires | Crown Fires | Confla-Grations | Down-Bursts | PyroEvs |
| Aberfeldy-Donnellys (expert 1) | 17/01/2013 | 86,781 | 1 | | 1 | | 1 | 1 | 1 | | 2 |
| Aberfeldy-Donnellys (expert 2) | 17/01/2013 | 86,781 | 1 | | 1 | | | | | | |
| Alpine | 1/04/2003 | 1,100,000 | 2 | | | | 2 | 2 | 2 | | 1 |
| Cann River-Myrgatroyd Track | 16/12/2009 | 6718 | | | | | | | 1 | | |
| Cape Conran-Dock Inlet | 2/11/2009 | 4199 | | | | | | | | | |
| Churchill-Jeeralang (expert 1) | 7/02/2009 | 1892 | 1 | 3 | | 3 | 1 | 1 | 1 | | 1 |
| Churchill-Jeeralang (expert 2) | 7/02/2009 | 1892 | 2 | 3 | | 3 | 1 | 1 | 1 | | 1 |
| Churchill-Jeeralang (expert 3) | 7/02/2009 | 1892 | 1 | | | | | | | | 1 |
| Club Terrace-Goolengook River | 16/01/2014 | 1863 | | | | | | | 1 | | |
| Dargo-Danes Track | 16/01/2014 | 1878 | | | | | | | 2 | | |
| Deep Lead | 31/12/2005 | 7506 | 1 | | | | | | 3 | | |
| Delburn | 29/01/2009 | 6474 | 2 | | | | | | 2 | | 2 |
| Genoa-Broome Creek Track | 16/01/2010 | 6319 | | | | | | | 1 | | 1 |
| Goongerah-Deddick Trail | 16/01/2014 | 2653 | 1 | | | | 1 | 1 | 1 | | 1 |
| Grampians-Northern Complex | 29/01/2014 | 54,590 | 1 | | | 2 | | | 1 | 1 | 1 |
| Grampians-Roses Gap Road | 22/01/2010 | 1715 | | | | | | | | | |

**Table A2.** *Cont.*

| Fire Name | Fire Date (yyyy/mm/dd) | Fire Area, Ha | Dynamic Fire Behaviours | | | | | | | | |
|---|---|---|---|---|---|---|---|---|---|---|---|
| | | | Spotting | Fire Tornado/Whirls | Fire Channelling | Junction Fires | Eruptive Fires | Crown Fires | Confla-Grations | Down-Bursts | PyroEvs |
| Grampians-Victoria Valley Complex (expert 1) | 1/01/2013 | 35,892 | 1 | | | | | | | 1 | 1 |
| Grampians-Victoria Valley Complex (expert 2) | 1/01/2013 | 35,892 | 1 | | | | | | | | |
| Grampians-Victoria Valley Complex (expert 3) | 1/01/2013 | 35,892 | | | | | | | | | 1 |
| Great Divide North | 1/12/2006 | 374,828 | 2 | | | | 2 | 2 | 2 | | 2 |
| Great Divide South | 1/12/2006 | 684,424 | 2 | | | | 2 | 2 | 2 | | |
| Harrietville-Alpine North | 21/01/2013 | 1129 | 2 | | | | 2 | 1 | 1 | | 1 |
| Harrietville-Alpine South | 7/02/2013 | 1149 | 2 | | | | 2 | 1 | 1 | | 1 |
| Horsham-Vectis | 7/02/2009 | 1782 | | | | | | | | | |
| Kal Kallo-Donnybrook Road | 18/02/2013 | 2042 | | | | | | | | | |
| Kinglake | 7/02/2009 | 123,455 | 2 | | | | 2 | 1 | 1 | | 1 |
| Little Desert-Broughtons Track | 3/01/2015 | 2848 | | | | | | | | | |
| Little Desert-Dahlenburgs Mill | 30/10/2009 | 1330 | | | | | | | | | |
| Little Desert-Lillimur | 8/02/2009 | 1066 | | | | | | | | | |
| Little Desert-Junkum Track | 7/12/2007 | 6440 | 1 | | | 1 | | | | | |
| Little Desert-Nhill Harrow Road | 7/01/2015 | 11,675 | 1 | | | 1 | | | | | 1 |
| Little Desert-Salt Lake Track | 18/11/2007 | 15,147 | | | | | | | | | |
| Little Desert-Tallageria Track | 31/12/2006 | 7987 | | | | | | | | 2 | |
| Little Desert-Wallaby Track | 3/01/2015 | 1075 | | | | | | | | | |
| Little Desert-Western Block Fire Complex | 1/01/2006 | 3555 | | | | | | | | | |
| Little Desesrt-Brooks Track | 20/11/2006 | 10,796 | | | | | | | | | |
| Mickleham-Kilmore | 9/02/2014 | 22,883 | | | | | | | | | |
| Marysville | 7/02/2009 | 132,874 | 2 | | | | 2 | 1 | 1 | | 1 |
| Mount Bolton-Laverys Road | 23/02/2016 | 1202 | 2 | | | | 2 | 1 | | | 1 |
| Moyston-Better Route Road | 2/01/2015 | 4455 | | | | | | | | | |
| Mt Clay-Golf Course Road | 1/04/2014 | 1249 | | | | | | | | | |
| Mt Lubra (expert 1) | 20/01/2006 | 127,137 | 1 | | 2 | 2 | 2 | 1 | 1 | | 1 |
| Mt Lubra (expert 2) | 20/01/2006 | 127,137 | 2 | | | | 2 | 1 | 1 | | 1 |
| Mt Lubra (expert 3) | 20/01/2006 | 127,137 | 1 | | | | | | | | |
| Mt Ray-Boundary Track | 16/01/2014 | 1336 | 2 | | | | 2 | 1 | | | 1 |
| Rocklands-Rees Road (expert 1) | 3/01/2015 | 4102 | | | | | | | | | |
| Rocklands-Rees Road (expert 2) | 3/01/2015 | 4102 | 1 | | | | 3 | | | 3 | 3 |
| Scotsburn Finns Road | 19/12/2015 | 4570 | | 1 | | | | | | | |
| Stawell- Bunjils Cave Road | 23/01/2014 | 1817 | | | | | | | | | |
| Timbarra-Gil Groggin | 8/02/2014 | 1971 | | | | | | | | | |
| Tostaree-Princes Hwy (expert 1) | 1/02/2011 | 10,992 | 1 | | | | 1 | 1 | | | 1 |
| Tostaree-Princes Hwy (expert 2) | 1/02/2011 | 10,992 | 2 | | | | 1 | 1 | | | 1 |
| Tostaree-Princes Hwy (expert 3) | 1/02/2011 | 10,992 | 1 | | | | | | | | |
| Wye River-Jamieson Track | 19/12/2015 | 2521 | 1 | | | | | | | | |

1 is direct measurements (linescans, images, video, etc.), 2 is indirect data (weather records, etc.) and 3 is the data based on anecdotal evidence (observations recorded in situation reports, etc.).

**Table A3.** South Australia.

| Fire Name | Fire Date (yyyy/mm/dd) | Fire Area, Ha | Dynamic Fire Behaviours | | | | | | | | |
|---|---|---|---|---|---|---|---|---|---|---|---|
| | | | Spotting | Fire Tornado/Whirls | Fire Channelling | Junction Fires | Eruptive Fires | Crown Fires | Confla-Grations | Down-Bursts | PyroEvs |
| Reef-Misery Tracks | 8/12/2015 | 1066 | | | | | | | | | 1 |
| Pinery | 25/11/2015 | 78,257 | | 1 | | | | | | | 1 |
| Kiana | 14/01/2014 | 6645 | | | | | | | | | |
| Sampson Flat | 2/01/2015 | 11,501 | 1 | | | | | | 1 | 2 | 1 |
| Bangor | 15/01/2014 | 33,207 | 1 | | 3 | 1 | 3 | | 3 | | 1 |
| Calperum | 14/01/2014 | 48,076 | | | | | | | | | 1 |
| Ngarkat | 14/01/2014 | 85,890 | | | | | | | | | 1 |
| Coomunga | 20/11/2012 | 1757 | | | | | | | | | |
| Billiatt | 14/01/2014 | 70,156 | | | | | | | | | 1 |
| Central | 6/12/2007 | 5032 | | | | | | 3 | 3 | | 3 |
| Destrees | 6/12/2007 | 16,285 | | | | | | 3 | 3 | | 3 |
| Chase | 6/12/2007 | 60,455 | | | | | | 3 | 3 | | 3 |
| Bookmark | 27/11/2006 | 118,356 | | | | | | | | | 1 |

1 is direct measurements (linescans, images, video, etc.), 2 is indirect data (weather records, etc.) and 3 is the data based on anecdotal evidence (observations recorded in situation reports, etc.).

**Table A4.** Tasmania.

| Fire Name | Fire Date (yyyy/mm/dd) | Fire Area, Ha | Dynamic Fire Behaviours | | | | | | | | |
|---|---|---|---|---|---|---|---|---|---|---|---|
| | | | Spotting | Fire Tornado/Whirls | Fire Channelling | Junction Fires | Eruptive Fires | Crown Fires | Confla-Grations | Down-Bursts | PyroEvs |
| Lake Macintosh | 25/01/2010 | 3395 | 3 | | | | | 3 | | | |
| Heemskirk Road | 15/03/2008 | 13,719 | 3 | | | | | 3 | | | |
| Asbestos Rd York Town | 2/01/2010 | 2926 | | | | | | | | | |
| Mount Castor | 20/01/2006 | 2699 | | | | | | | | | |
| Meadowbank Road | 25/02/2012 | 5234 | 3 | | | | | 3 | | | |
| Wayatinah | 31/01/2010 | 6285 | 3 | | | | | 3 | | | |
| Trial Harbour Rd | 20/01/2006 | 3209 | | | | | | | | | |
| Elliot Bay | 20/01/2006 | 1296 | | | | | | | | | |
| Valley Road Fingal | 6/02/2013 | 2039 | | | | | | | | | |
| Giblin River | 3/01/2013 | 40,463 | 3 | | | | | 3 | 3 | | |
| Lake Repulse | 3/01/2013 | 10,238 | 3 | | | | | 3 | | | |
| Inala Road-Forcett | 3/01/2013 | 23,362 | 3 | | | | | 3 | 3 | | 2 |
| Granville Harbour Rd | 29/11/2012 | 1639 | | | | | | | | | |
| Ainslie Beach | 2/03/2013 | 1118 | | | | | | | | | |

**Table A4.** *Cont.*

| Fire Name | Fire Date (yyyy/mm/dd) | Fire Area, Ha | Dynamic Fire Behaviours | | | | | | | | |
|---|---|---|---|---|---|---|---|---|---|---|---|
| | | | Spotting | Fire Tornado/Whirls | Fire Channelling | Junction Fires | Eruptive Fires | Crown Fires | Confla-Grations | Down-Bursts | PyroEvs |
| Poatina | 29/11/2012 | 8512 | | | | | | 3 | | | |
| Glen Dhu Road | 22/02/2013 | 2612 | 3 | | | | | 3 | | | |
| Zeehan Hwy | 11/11/2012 | 1398 | | | | | | | | | |
| Surprise Creek | 17/01/2014 | 2426 | | | | | | | | | |
| Top Farm Track Granville | 14/03/2011 | 1549 | | | | | | | | | |
| Banca Road | 3/11/2013 | 1682 | 3 | | | | | 3 | | | |
| Lake Burbury | 27/01/2014 | 2114 | 3 | | | | | 3 | | | |
| Curries Dam Road | 3/03/2015 | 1061 | | | | | | 3 | | | |
| Rebecca | 14/10/2014 | 4833 | 3 | | | | | 3 | | | |

1 is direct measurements (linescans, images, video, etc.), 2 is indirect data (weather records, etc.) and 3 is the data based on anecdotal evidence (observations recorded in situation reports, etc.).

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
