# Peer review of "Frequency of Dynamic Fire Behaviours in Australian Forest Environments"

_fire, doi:10.3390/fire3010001_

Round 1
Reviewer 1 Report
This article describes the results obtained from a survey about what the authors call “Dynamic Fire Behaviours (DFBs)”. The survey collected the input from 9 fire experts in Australia about their experience with fire behaviour patterns that can potentially lead to extreme fire events.
The main alleged contributions of this article can be listed as: 1) proposal of the term DFB to refer to sudden changes in fire behaviour that can lead to extreme fire behaviour: 2) estimation of the relative frequency of occurrence of various phenomena considered within this category. I do not think that either of these contributions is achieved successfully in this paper and I have strong concerns about the suitability of this article for publication, as described below.
Firstly, the authors seem to propose the use of the new term DFB to refer to fire behaviour patterns such as crown fire (no distinction is made between active and passive crown fire), spotting, fire whirls and pyrocumulonimbus. However, the introduction is misleading and it does not provide a clear idea of the state of the art and the authors’ proposal about the use of a new term. A number of studies are cited in the introduction when describing previous work about DFB. However, most of these original works do not refer to these phenomena as DFB but as extreme fire behaviour. This is not clearly stated in the paper. The manner in which the introduction is written suggests that there is significant previous literature about DFB, when this is not the case. Conversely, the authors later on clearly state that they suggest replacing the use of “extreme fire behaviour” with “dynamic fire behaviours”. I find the complete framework of this proposal rather confusing. The need for a new term of this type is highly questionable. If the authors do intend to propose the use of a new term, there should be a very strong justification and clear presentation of why this is convenient or necessary. Specifically about the term DFB, I don’t find it appropriate. Fire is a dynamic physical process. It may be steady or unsteady, but referring to rapid changes in fire behaviour as DFB does not make sense from a physical point of view.
Secondly, the methodology of the statistical study described in section 2 is invalid. The authors relate population and sample size to the body of experts who reported data, whereas the actual study involves fire, not the experts. When applied to fires instead of experts, the sample size estimation tool used by the authors (https://www.abs.gov.au/websitedbs/D3310114.nsf/home/Sample+Size+Calculator) recommends using a sample size 273 when the population size is 934 in order to achieve 95% confidence. The authors refer using a list of 934 fires for surveying. From that target population, they received data for only 113 samples, which is far below the recommended sample size. Furthermore, data collected from 9 experts mainly concentrated in a single region may not be representative of general trends. I also found percentages given throughout the paper confusing and misleading. Finally, no details are given about the job positions of the experts or the data they used to identify the different DFBs.
The subject of this paper is relevant to this journal and the existence of a database with information about the frequency of extreme fire behaviour patterns would be highly valuable for the community. However, I believe that neither the methodology not the results described in this paper meet the standards of a scientific publication. There is no useful conclusion that can be drawn from this study. Therefore, I recommend the authors revise their methodology, increase their sample size and re-do this study more rigorously, potentially including information about the conditions that led to each observed phenomenon.
Author Response
Please find all answers in the attachment.

Reviewer 2 Report
Summary and contribution
The authors surveyed fire and land management staff about dynamic fire behaviors (DFBs), which include spotting, crown fires, pyro-convective events, eruptive fires, conflagrations, junction fires, fire tornados/whirls, fire channelling and downbursts. In the introduction, the authors define DFB, a new term, and describe how DBFs relate to extreme fire events or behavior. Eleven people responded to their inquiries about a total of 934 fires. The authors analyzed DFB frequency and type, and found that DFBs occurred in 60% of fires, DFBs occurred more often in larger fires (although not significant at p=0.05), and that 73% of fires had multiple DFBs, of which spotting, crown fires, and pyro-convective events were most common.
Unpredictable fire behavior such as DFBs pose a risk to human safety and lives. This paper analyzes DFBs to better understand DFB frequency by type, where little or no previous work has been done. As such, this paper is a valuable contribution to the better understanding of unpredictable fire behavior, and merits publication.
Major comments
The paper is well written and organized, and easy to read, and contains valuable information. The figures are effective.
One weakness of the analysis is the low number of respondents, eleven (hence my rating of ‘can be improved’ for the research design and my rating of ‘low’ for scientific soundness; more respondents would be better obviously). However, the authors did a good job with the data they did get and even working with 11 respondents' data required a large research effort I imagine. Also, the authors are clear about the qualitative nature of the analysis, that some of the observations are indirect and anecdotal, and suggest more quantitative analyses in the future.
Another weakness is that the most common DFBs identified in the analysis were also those that were most easily identified; the less common DFBs might occur more frequently than this analysis suggests. However, the authors state this fact in the discussion so the reader is aware. Despite these weaknesses, I think the analysis still has value and should be published.
As I point out in minor comments below, I disagree with some of the authors’ interpretations of results. In my opinion, relationships do seem to exist between fire size and DFB occurrence, as well as the number of DFBs per fire and fire size (hence my rating of ‘can be improved’ for the question of whether the conclusions are supported by the results).
Minor comments
Lines 161, 162: Although not significant at p=0.05, the relationship was significant at p=0.1. So I would interpret the results differently. In my opinion, there was a relationship, albeit a weak and nonsignificant relationship. The authors could simply state that there was no significant relationship.
Line 208, 209: A correlation coefficient of 0.25 and a p-value of 0.014 suggests that there was a significant relationship between the number of DFBs per fire and fire size. Perhaps the reported p-value is written incorrectly? A p-value < 0.05 is considered significant by many standards. But like my previous comment, and after looking at Figure 3a, there looks to be a weak relationship between the two.
Author Response
Please find our answers in the attachment.

Reviewer 3 Report
General comment
This manuscript (MS) presents an analysis of dynamic forest fire behaviour in Australian wildfires.
The work is concise and interesting. Namely, I find the structure, contents and MS length are well suited to the type of paper the authors wish to publish, a ‘research note’. I have only minor comments, that I hope can help authors in further improving the MS.
Specific comments
Abstract:
- Lns 14-16: The sentence is difficult to read. Please revise.
- Lns 16-17: The sentence is difficult to understand. Do you mean that one or more DFBs where observe in 60% of the 113 fires? Please rewrite for clarity.
- Lns 21-23: The sentence is confusing. Perhaps you mean ‘be’ instead of ‘for’?
Introduction:
- Lns 27-28: Please revise for clarity.
- Lns 32-33: There seems to be a lack of agreement between the verbs. Do you mean ‘could burn’ or perhaps ‘result in human losses’?
- Ln 35: Perhaps ‘loss of lives’?
- Lns 35-36: In 2017, 112 people died in Portugal as a result of two major wildfires, both in Central Portugal, one in Jun-17 and the other in Oct-15.
- Ln 38: What do you mean by ‘which occur within any fires’? Do you mean that they occur frequently or that they always occur in a fire?
- Ln 40: The part of the sentence ‘of the nature’ does not seem necessary. Also, although there might be a limited understanding of the drivers of extreme fire behaviour, there is evidence on the conditions under which most of these events occur. You should add some text on this.
Frequently, during the fire season, hot and dry weather is observed. This is a very common event which originates low dead fuel moisture content and facilitates fire spread. But the atypical most extreme fire events are usually observed when fires spread under a combination of high wind speeds and drought conditions. This is because live fuel moisture content (LFMC) reaches very low values, as a result of severe water deficit. Usually, live fuels require the presence of dead fuels to support fire spread, but when LFMC goes below 60% (Rossa and Fernandes 2018, doi:10.1093/forsci/fxy002) live fuels are able to spread fire by themselves. Extremely low LFMC has a dramatic impact fire behaviour in two ways: i) very low overall FMC (live and dead fuels) enhances fire-spread rate and makes a huge amount of fuel load available for combustion, ii) landscape connectivity is increased because live fuels can support fire-spread in the absence of dead fuels. The two references below, recently published in Fire, address the effect of wind on fire spread and the subject of LFMC:
Rossa CG, Fernandes PM (2018) Live fuel moisture content: The ‘pea under the mattress’ of fire spread rate modeling? Fire 1(3), 43. doi:10.3390/fire1030043
Materials and methods:
- Lns 91-93: Why are the first letters of the vegetation types capitalized?
Results and discussion:
- Ln 160: In the caption of Figure 1, DBFs should be written in full with the abbreviation in parenthesis: Dynamic Fire Behaviours (DBFs).
- Lns 175-177: In the caption of Figure 2, in its first appearance DBFs should be written in full with the abbreviation in parenthesis: Dynamic Fire Behaviours (DBFs).
- Lns 206-207: In the caption of Figure 3, in its first appearance DBFs should be written in full with the abbreviation in parenthesis: Dynamic Fire Behaviours (DBFs). Also, the meaning of the abbreviations in Figure 3a ‘VIC, NSW, etc.’ should be defined in the caption.
- Lns 215-217: The meaning of the abbreviations in Figure 4b ‘VIC, NSW, etc.’ should be defined in the caption.
Author Response

(The authors gave the same response as above.)

Reviewer 4 Report
Very interesting research about dynamic fire behaviours. The research note is relevant to the aims and scope of the journal. The title is concise, the introduction is well documented, the material and methods are clear, the results and discussion are well explored.
Author Response
Thank you for positive evaluation of our work!
Round 2
Reviewer 1 Report
I am slightly disappointed by the approach taken by the authors in their response to reviews in this article. Instead of addressing the concerns raised by reviewers, the authors rather focused on refuting reviewer comments because reviewers apparently “missed the point”.
After going through the revised version of this article, I still cannot recommend its publication. Most of my concerns have not been addressed. In particular:
1) The first important point made by the authors is the proposal of the term “Dynamic Fire Behaviours (DFBs)” as a replacement for Extreme Fire Behaviour. However, this proposal is not clearly stated nor justified in the text. I am especially concerned about the use of references in the introduction. Hopefully unintentionally, the authors include incorrect quotes in lines 60-65. In an attempt to justify the use of “DFB”, a series of articles are cited in which DFB were previously defined, according to the authors. However, none of those authors mention the term “DFB” in their manuscripts, but they instead use “Extreme Fire Behaviour”.
In my opinion, the correct approach would be to review existing literature about Extreme Fire Behaviour, explain why the term “Extreme” is incorrect based on available knowledge and justify the alternative use of DFB. I believe that the current introduction of this paper is misleading for the reader and unfair to referenced authors. I raised this concern in my previous review and it has not been addressed.
2) The second significant contribution of this paper would be the collection of real-scale wildfire observations acquired indirectly through emergency managers. While these data can be very valuable for the community, I don’t believe the approach is clear in this paper.
In their response to my previous comments as well as in lines 253-254 of the revised manuscript, the authors state that they could not collect enough data for an exhaustive quantitative analysis. This is completely understandable and I still believe that this data is useful. However, the paper is full of percentages and statistical statements. If data is not enough for a quantitative study, statistical analyses are not reliable. A few global percentages may be useful to present the relative frequency of different phenomena and discuss where further research should be focused, but I believe that statistics are largely overused in this article. They are even contradictory sometimes. For example, the abstract says that sixty percent of the identified fires had one or more DFBs, whereas lines 211-212 state that it was 73% of the fires that “two and more DFBs were recorded” in. Moreover, 73% of the sample fires is approximately 82.5. That implies that there were more fires with at least two DFBs than fires with one or more DFBs (line 211). Line 211 refers to figure 2b but figure 2b contains no information related to these percentages.
In summary, I find the tone of this paper overstated and I don’t think that enough effort has been put into the preparation of the article itself. While the performed study and the collected data may be useful for the community, I can only suggest that the authors re-write their paper and submit it again in a different shape. If I were the authors, I would consider splitting the article and treating the two following topics separately: 1) proposal of the use of DFB instead of Extreme Fire Behaviour based on a strong and exhaustive literature review: and 2) general conclusions about the relative frequency of various types/categories of fire behaviour phenomena, based on data collected indirectly through fire emergency managers.
Author Response
I am slightly disappointed by the approach taken by the authors in their response to reviews in this article. Instead of addressing the concerns raised by reviewers, the authors rather focused on refuting reviewer comments because reviewers apparently “missed the point”. After going through the revised version of this article, I still cannot recommend its publication. Most of my concerns have not been addressed. In particular:
1) The first important point made by the authors is the proposal of the term “Dynamic Fire Behaviours (DFBs)” as a replacement for Extreme Fire Behaviour. However, this proposal is not clearly stated nor justified in the text. I am especially concerned about the use of references in the introduction. Hopefully unintentionally, the authors include incorrect quotes in lines 60-65. In an attempt to justify the use of “DFB”, a series of articles are cited in which DFB were previously defined, according to the authors. However, none of those authors mention the term “DFB” in their manuscripts, but they instead use “Extreme Fire Behaviour”. In my opinion, the correct approach would be to review existing literature about Extreme Fire Behaviour, explain why the term “Extreme” is incorrect based on available knowledge and justify the alternative use of DFB. I believe that the current introduction of this paper is misleading for the reader and unfair to referenced authors. I raised this concern in my previous review and it has not been addressed.
It is apparent that there is ambiguity in our writing here. We have attempted to address the reviewers concerns here. The introduction now outlines the arguments made regarding what has previously been termed extreme fire behaviours. We then argue why the definition is no longer appropriate and propose the new term dynamic fire behaviour. We trust this now addresses the concerns of the associate editor and the reviewer.
2) The second significant contribution of this paper would be the collection of real-scale wildfire observations acquired indirectly through emergency managers. While these data can be very valuable for the community, I don’t believe the approach is clear in this paper.
In their response to my previous comments as well as in lines 253-254 of the revised manuscript, the authors state that they could not collect enough data for an exhaustive quantitative analysis. This is completely understandable and I still believe that this data is useful. However, the paper is full of percentages and statistical statements. If data is not enough for a quantitative study, statistical analyses are not reliable. A few global percentages may be useful to present the relative frequency of different phenomena and discuss where further research should be focused, but I believe that statistics are largely overused in this article.
We disagree with the reviewer here. Most of the paper is based on percentage of observations and the reviewers agree this is appropriate. The only other analysis we conducted was to analyse the relationships with fire sizes. These analyses are based on 113 fires. Given these are univariate analyses, our data exceeds the minimum sample size for such analyses. The terminology in lines 253-254 was awkward and we have modified it:“It should be noted that the obtained values are not the definitive values, as the limited amount of data restrict quantitative analysis. Several attempts were made to increase the database, but despite sustained efforts we were unable to increase it substantially. However, obtained values provide useful insights into frequencies of DFBs and can be used for the prioritization of future research.”
They are even contradictory sometimes. For example, the abstract says that sixty percent of the identified fires had one or more DFBs, whereas lines 211-212 state that it was 73% of the fires that “two and more DFBs were recorded” in. Moreover, 73% of the sample fires is approximately 82.5. That implies that there were more fires with at least two DFBs than fires with one or more DFBs (line 211). Line 211 refers to figure 2b but figure 2b contains no information related to these percentages.
We appreciate the sample size for the percentage values were not always clear. We have added text throughout to make it clear what sample size we are talking about.
In summary, I find the tone of this paper overstated and I don’t think that enough effort has been put into the preparation of the article itself. While the performed study and the collected data may be useful for the community, I can only suggest that the authors re-write their paper and submit it again in a different shape. If I were the authors, I would consider splitting the article and treating the two following topics separately: 1) proposal of the use of DFB instead of Extreme Fire Behaviour based on a strong and exhaustive literature review: and 2) general conclusions about the relative frequency of various types/categories of fire behaviour phenomena, based on data collected indirectly through fire emergency managers.
We appreciate the reviewer’s comments but find these contradictory. On one hand they say we are overstating the results and on the other they are advocating two papers. Given the positive comments of three of the original reviewers and the revisions made in response to these comments we hope the paper is considered suitable for publication.